# Photoluminescent Gold/BSA Nanoclusters (AuNC@BSA) as Sensors for Red-Fluorescence Detection of Mycotoxins

**DOI:** 10.3390/ma15238448

**Published:** 2022-11-27

**Authors:** Ivana Fabijanić, Marta Jurković, Daniela Jakšić, Ivo Piantanida

**Affiliations:** 1Division of Organic Chemistry and Biochemistry, Laboratory for Biomolecular Interactions and Spectroscopy, Ruđer Bošković Institute, 10002 Zagreb, Croatia; 2Department of Microbiology, Faculty of Pharmacy and Biochemistry, University of Zagreb, 10000 Zagreb, Croatia

**Keywords:** AuNC@BSA nanoclusters, mycotoxins, fluorescence, non-covalent binding

## Abstract

The BSA-encapsulated gold nanoclusters (AuNC@BSA) have drawn considerable interest and demonstrated applications as biological sensors. In this study, we demonstrated that the red-emitting AuNC@BSA prepared using a modified procedure fully retained the binding of standard BSA-ligands (small molecule drugs), significantly improving fluorescence detection in some cases due to the red-emission property. Further, we showed that AuNC@BSA efficiently bind a series of aflatoxin-related mycotoxins as well as the aliphatic mycotoxin FB_1_, reporting interactions in the nanomolar range by instantaneous emission change at 680 nm. Such red emission detection is advantageous over current detection strategies for the same mycotoxins, based on complex mass spectrometry procedures or, eventually (upon chemical modification of the mycotoxin), by fluorescence detection in the UV range (<400 nm). The later technique yields fluorescence strongly overlapping with the intrinsic absorption and emission of biorelevant mixtures in which mycotoxins appear. Thus, here we present a new approach using the AuNC@BSA red fluorescence reporter for mycotoxins as a fast, cheap, and simple detection technique that offers significant advantages over currently available methods.

## 1. Introduction

In the last few decades, there has been a growing interest in nanoparticles with tailored properties on demand, with an emphasis on the development of simple, inexpensive, and eco-friendly synthetic protocols under mild experimental conditions, using mostly water as a solvent and weak reducing agent. So far, a big step forward has been made in tuning the properties of metal nanoparticles by tuning their size, shape, or coating [1,2,3,4]. Metal nanoclusters (mostly gold [5,6], silver [7], or bimetallic [8]) protected by different biomolecules, such as bovine serum albumin (BSA) [4,5], human serum albumin (HSA) [9], deoxyribonucleic acid (DNA) [10], or horseradish peroxidase (HRP) [11], certainly stand out as a new generation of nanoparticles. Among them, BSA-encapsulated AuNCs (AuNC@BSA) have already shown great potential both in biomedical/nanomedical research and in practical application due to their unique properties [12,13,14]. Their ultra-small size, unique molecular-like behavior, such as tunable emission defined by the number of atoms forming the nanocluster [5,15], simplicity of one-pot synthesis, biocompatibility, negligible cytotoxicity [14], stability [4], and relatively long photoluminescence lifetime [16], make AuNC@BSA a perfect tool for various applications in biotechnology. To date, AuNC@BSA nanoclusters have been successfully used as a fluorescent probe for selective ion detection, i.e., chlorine [11], hypochlorite [17], copper, and mercury [18], but also for the detection of many organic compounds, such as haemoglobin [19], flavonoids [20], or drugs [21]. Because of their tuneable enzyme-like feature, AuNC@BSA could be used as catalysts, as they are enzymatically active under milder, more biologically acceptable conditions compared to enzymatic reactions of natural enzymes [5,19], and due to their ability to create various organic molecule-cluster hybrids or nanocomplexes, they could be used for gene delivery [22], fluorescence imaging, and targeted drug delivery [23].

This study focuses on mycotoxins, which are naturally occurring toxic secondary metabolites of various fungi that are widely present in the environment. There are several hundred different mycotoxins, but the most important are aflatoxins, ochratoxin A, patulin, fumonisins [24], zearalenone, and nivalenol/deoxynivalenol, produced by fungi from the genera *Aspergillus*, *Penicillium,* and *Fusarium* because of their harmful impact on every part of the food chain, i.e., plants, animals, and humans. The document issued by the European Food Safety Authority in 2013 [25] pointed to insufficient information on the occurrence of sterigmatocystin (STC) in food matrices, a prerequisite for characterising its risk for human health, and the need for the development of more sensitive methods for detection of STC. Therefore, in the past decade, there has been a growing research interest in STC. Even though STC is considered less potent than its sibling mycotoxin, aflatoxin B_1_ (AFB_1_), people may be more exposed to STC than AFB_1_ because it is one of the most frequent mycotoxins in indoor environments [26,27,28]. However, people may be exposed to the mycotoxins not only by ingestion of contaminated food but also by inhalation or dermal exposure. In this regard, STC and its methoxy derivative, 5-methoxysterigmatocystin (5-OMe-STC), are particularly interesting because of their high prevalence in indoor living and working environments [24]. 

The interactions between mycotoxins and HSA are studied as biomarkers of exposure [29]. The most thoroughly studied and best-known among the mycotoxins is certainly AFB_1,_ a group 1 carcinogen [30,31]. The exposure assessment study showed that 1.4–2.3% of ingested AFB_1_ forms covalent adducts with serum albumin [32]. Thus, in the design of this work, we relied on the well-documented binding of mycotoxins to serum albumin and proposed that AuNC@BSA nanoclusters would retain this mycotoxins-binding property, reporting the interaction by the change in red-emitting AuNC@BSA emission (Figure 1).

In that way, new sensitive and red-emitting fluorimetric sensors for mycotoxins would be available for point-of-care applications.

## 2. Materials and Methods

### 2.1. Materials

Tetrachloroauric(III) acid trihydrate (HAuCl_4_·3H_2_O, ≥99.9% trace metal basis), sodium hydroxide (NaOH, ASC reagent, ≥97%), bovine serum albumin (lyophilized powder BSA, ≥96%, agarose gel electrophoresis, BioReagent), ethanol (absolute for analysis, EMSURE^®^ ACS, ISO, Reag. Ph Eur), and AFB_1_ (from *Aspergillus flavus*) were purchased from Sigma Aldrich (St. Louis, MO, USA). STC (C_12_H_18_O_6_, ≥98%, from *A. versicolor*) and FB_1_ (≥95%, crystalline solid) were purchased from Cayman Chemical Company (Ann Arbor, MI, USA). Abcam (Boston, MA, USA) supplied the 5-OMe-STC > 98% solid, from *Aspergillus* sp.

Meloxicam (MEL, C_14_H_13_N_3_O_4_S_2_, >96.0%) and diclofenac sodium salt (DSS, C_14_H_10_Cl_2_NNaO_2_, >98.0%) were purchased from Tokyo Chemical Industry (TCI, Tokyo, Japan); Dimethyl sulfoxide (DMSO, spectrophotometric grade, >99.9%); and acetonitrile (HPLC grade, ≥99.7%) were purchased from Alfa Aesar (Lancashire, UK). All chemicals were used without further purification. For all experiments, ultrapure water was used. G-Biosciences (St. Louis, MO, USA) supplied the Tube-O-Dialyzer^TM^ (Medi, 4 kDa MWCO).

UV-Vis measurements were performed on a Varian Cary 100 Bio (Agilent Technologies, Vienna, Austria), and fluorescence measurements were performed on a Cary Eclipse Fluorescence Spectrophotometer (Agilent Technologies, Vienna, Austria) and an Edinburgh Instruments FS5 Spectrofluorimeter equipped with Fluoracle^®^ software (Edinburgh Instruments Ltd., Livingston, UK). The resulting titration data were fitted with an exponential function, obtaining *R*^2^ > 0.99 in all cases. TC-SPC measurements were also performed on an Edinburgh FS5 spectrofluorometer equipped (Edinburgh Instruments Ltd., Livingston, UK) with a pulsed LED at 340 and 560 nm. The duration of the pulse was 500–1000 ps. Fluorescence signals at 680 nm were monitored over 1023 channels with a time increment of 5–20 µs/1024 channels. The decays were collected until they reached 3000 counts in the peak channel. A suspension of silica gel in H_2_O was used as a scattering solution to obtain the instrument response function (IRF). All measurements were performed in 10 mm × 10 mm cuvettes at room temperature (25 °C). Decays of fluorescence were fitted using a multi-exponential fitting model (1): (1)R(t)=∑iBie−tτi 
where *B_i_* is the amplitude of the decay of the *i-*th component at time *t*, and *τ* is the lifetime of the *i*-th component. The visualisation of nanoclusters was performed using a transmission electron microscope, JEM1010 (Jeol Ltd., Tokyo, Japan), operated in bright field mode at an acceleration voltage of 80 kV. Images were recorded with a MegaView CCD camera (Olympus, Tokyo, Japan), attached to the microscope. A TEM sample was prepared by depositing a drop of nanocluster suspension onto a formvar-coated copper grid (200HX, 200 Mesh, SPI Supplies (Philadelphia, PA, USA) and air-drying at room temperature. TEM images were analyzed using ImageJ software [34], and the average diameter of the nanoclusters was obtained from the analysis.

### 2.2. Synthesis of AuNC@BSA

AuNC@BSA were synthesised following the procedure from Xie et al. [4] with minor modifications. In a typical experiment, all glassware used in the experiments was cleaned with freshly prepared aqua regia (HCl: HNO_3_ volume ratio = 3:1), rinsed thoroughly in ultrapure water, and dried with nitrogen before use. Then, 5 mL of 10 mM water solution of HAuCl_4_ × 3H_2_O (thermostated at 40 °C for 30 min, protected from light) was added to 5 mL of a 50 mg/mL aqueous BSA solution (thermostated at 40 °C for 30 min, protected from light) under vigorous stirring (600 rpm). After 5 min of stirring, 500 µL of 1M NaOH was added to the reaction mixture. After NaOH addition, the reaction mixture was incubated in the water bath, protected from light, at 40 °C for 24 h. The product was kept in the fridge, protected from light, and used within one month. For lifetime measurements, AuNC and BSA were dialyzed for 48 h in 500 mL of ultrapure water using 4 kDa MWCO dialysis tubing (at least 8 times water exchange). The concentration of as-synthesised AuNC@BSA could be expressed as the total molar concentration of BSA in the reaction (*c*(BSA, AuNC@BSA) = 358 μM). Or, if we assume the completeness of the reaction and that each nanocluster contains 25 gold atoms [4], then the concentration of AuNC@BSA can be calculated from the concentration of the gold precursor (*c*(Au_25_, AuNC@BSA) = 190 μM) [35].

### 2.3. Spectrophotometric Study of AuNC@BSA Interactions with Ligands

To determine whether BSA in the nanocluster retained its binding properties, fluorimetric titrations of AuNC@BSA with two model molecules, meloxicam (MEL) and diclofenac sodium salt (DSS) [36], were performed. These molecules bind well to BSA, and their binding constants to BSA have been previously published [37,38]. After it was proven that BSA retained its ligand-binding properties in the nanocluster, the binding constants of AuNC@BSA with four mycotoxins—AFB_1_, STC, 5-OMe-STC, and FB_1_—were determined. All measurements were performed in the same way and under the same experimental conditions, as follows: (A) The AuNC@BSA fluorescence probe was first prepared by adding 4 μL of as-synthesized AuNC@BSA stock suspension in 2000 μL of ultrapure water, and then (B) aliquots of freshly prepared model molecule/mycotoxin solutions were added to the previously prepared AuNC@BSA fluorescence probe. After 15 min of incubation at 25 °C, emission spectra were recorded under excitation at 540 nm after each aliquot addition. Titration data were processed by a non-linear fitting procedure, e.g., multivariate analysis using the SPECFIT GLOBAL ANALYSIS, A Program for Fitting, Equilibrium and Kinetic Systems, using Factor Analysis & Marquardt Minimization [39] for various possible stoichiometries of complexes that could be formed. 

## 3. Results and Discussion

### 3.1. Characterisation of AuNC@BSA

The preparation of AuNC@BSA is a process sensitive to experimental conditions, and we have somewhat modified the synthetic procedure; therefore, it was essential to fully characterise the prepared AuNC@BSA. Thus, synthesised AuNC@BSA has been characterised by UV-Vis, fluorescence-spectroscopy, and TEM measurements. 

The TEM microscopy (Figure 1a) showed the typical pattern of rather homogeneous nanoparticles. The average diameter of AuNC@BSA is 2.8 ± 0.2 nm (Appendix A), which is in correspondence to the previously reported results for red-shifted nanoclusters [5].

The prepared AuNC@BSA stock solution showed typical red colour and luminescence (Figure 1b). More detailed analysis by fluorimetric methods performed in an aqueous solution showed fluorescence emission and excitation spectra (Figure 1c), agreeing well with previously reported data [4]. When excited at 365 nm, AuNC@BSA emit two characteristic emission peaks at 431 nm and 680 nm, originating from BSA and AuNC, respectively [40]. AuNC@BSA emits only one emission peak at 680 nm, originating only from AuNC when nanoclusters are excited at 540 nm. The UV-Vis properties (Appendix A) of AuNC@BSA agree well with the literature data [4], showing absorbance of BSA below 300 nm and an absorption shoulder (noticeable also in the excitation spectrum) between 450 and 650 nm corresponding to AuNC (Appendix A).

Since the fluorescence emission of AuNC@BSA is an essential tool in this study, we also characterised the emission lifetime decay properties at different pH values (Appendix A) and on two different excitation wavelengths (λ_exc_ = 339 and 560 nm). The obtained values at pH 12 are within the same order of magnitude as previously published data [15].

### 3.2. Interactions of AuNC@BSA with Meloxicam (MEL) and Diclofenac Sodium Salt (DSS)

To determine whether BSA in the AuNC@BSA nanocluster retained its ligand-binding properties, fluorimetric titrations were performed with MEL and DSS, two non-steroidal anti-inflammatory drugs known to bind well to BSA. Seedher and Bhatia [37] reported a binding constant of 6.548 × 10^5^ M^−1^ at 293.15 K and pH = 7.4 for a 1:1 interaction stoichiometry between BSA and MEL in 0.1 M phosphate buffer, determined by the fluorescence quenching of BSA after excitation at 296 nm. The measured data were corrected for the inner filter effect. Figure 2a shows the fluorimetric titration of AuNC@BSA with MEL in water at 298 K. The obtained binding constant for the interaction of AuNC@BSA with MEL is 3.73 × 10^5^ M^−1^, agreeing well with the previously published value [37].

However, since MEL has no absorbance at the excitation wavelength of AuNC@BSA, no inner filter effect corrections were needed, and thus the biding constant could be more accurate (Figure 2b). It should be noted that the fluorescence of AuNC@BSA is only about 50% quenched when reaching the full saturation plateau, thus complexes are still measurably emissive. 

The binding constants of BSA with DSS were previously determined with the equilibrium dialysis method [38]. According to the authors, it appears that the interaction of BSA with DSS is a two-site binding process, with binding constants *K*_ass1_ = 6.00 × 10^5^ M^−1^ and *K*_ass2_ = 1.59 × 10^3^ M^−1^ at 293.15 K, pH = 0.4, and *I* = 0.1 M. Figure 3a,b show the fluorometric titration of AuNC@BSA with DSS in water at 298 K with λ_exc_ = 365 nm and λ_exc_ = 540 nm, also supporting the formation of two different complexes—however, small emission changes prevented accurate calculation of binding constants.

Analogous fluorometric titration was performed with λ_exc_ = 540 nm, and titration data were analysed by multivariate analysis using the SPECFIT programme [39] for various possible stoichiometries of complexes that could be formed. 

The only acceptable agreement between experimental and calculated data (Figure 3d–f) was obtained with the model of two complexes: AuNC@BSA/DSS = 1:1 and AuNC@BSA/DSS = 1:2. The obtained cumulative binding constants *β*_11;12_ = 4.2 × 10^7^ ± 1.1 M^−2^ and *K*_11_ = 6.3 × 10^5^ ± 0.9 M^−1^; since *β*_12_ = *K*_11_ × *K*_12_; then *K*_12_ = 660 M^−1^; thus, the obtained results agree very well with the literature data [38].

All fluorimetric titration experiments were repeated several times, searching for optimal conditions, and RSD values (RSD < 10%) obtained for the binding affinity constants demonstrated that this is a sensitive and reliable method for studying BSA-ligand systems.

### 3.3. Detection of Mycotoxins

After determining that our AuNC@BSA retained its full capacity of binding BSA-targeting small molecules, we further studied the interactions of AuNC@BSA with a chosen set of mycotoxins (Figure 1). The addition of any of these mycotoxins resulted in a pronounced quenching of AuNC@BSA emission (Figure 4), similar to the effect caused by MEL (Figure 2) and more pronounced than noted for DSS (Figure 3).

Analysis of titration data (Figure 4a–d) by multivariate analysis using the SPECFIT program [39] gave the best fit for 1:1 stoichiometry complex formation (Figure 4e) and yielded binding constants (Table 1).

The binding constants obtained were mostly in the same order of magnitude as those obtained for the reference ligands MEL and DSS (Table 1), with only AFB_1_ affinity being significantly stronger. 

When compared to the literature data, the HSA/AFB_1_ complex binding constant (*K_a_* = 6.02 × 10^4^ M^−1^) [41] was found to be much lower, requiring authors to correct titration data for the inner-filter effect in a concentration-dependent fashion, a laborious mathematical modulation of experimental data prone to systematical deviation. Since our approach does not have to be corrected for the inner-filter effect, it allows for a more simple and accurate measurement and very likely a more accurate determination of binding affinity.

Further, there was no literature data about STC/serum albumin affinity; therefore, we determined the binding constant of STC to HSA by direct fluorimetric titration (Appendix A), which yielded a binding constant *K =* 20 × 10^5^ M^−1^, somewhat higher than obtained for AuNC@BSA (Table 1), but again, it has to be considered that direct titration (Appendix A) is performed under possible inner-filter effect conditions.

Additionally, to the best of our knowledge, there is no literature data for FB_1_/serum albumin affinity. Therefore, we performed titration of native BSA with FB_1_ (Appendix A), taking advantage of intrinsic BSA fluorescence, whereby FB_1_ does not absorb light within the protein excitation range—thus, we did not require correction for the inner filter effect. The obtained binding constant *K =* 8.9 ± 1.1 × 10^5^ M^−1^ was higher than that determined for AuNC@BSA (Table 1), however, within the same order of magnitude.

## 4. Conclusions

In this paper, we demonstrate that red-fluorescent AuNC@BSA can be used as a fluorescent probe for the detection of natural mycotoxins. Experimental results show that 4 mycotoxins—STC, 5-OMe-STC, AFB_1_, and FB_1_—interact with BSA in AuNC@BSA at nanomolar concentrations. Fluorescence quenching of AuNC@BSA at 700 nm after excitation at 540 nm was used to determine the binding constants of mycotoxin/AuNC@BSA. 

Current detection strategies for these mycotoxins are based on complex mass spectrometry procedures or eventually (upon chemical modification of the mycotoxin) by fluorescence detection in the UV range (<400 nm). However, fluorescence detection sometimes includes a substantial overlap of the fluorescence of the modified mycotoxin with the fluorescence of many fluorophores in biorelevant mixtures. Therefore, it requires mathematical modulation of the experimental data after correction of the concentration titration data for the inner-filter effect. Our method provides direct calculation of mycotoxin’s concentration from fluorescence quenching of AuNC@BSA without chemically modifying mycotoxin or mathematical modulation of experimental data, finally obtaining more accurate results. The experimentally obtained binding constants of mycotoxins with AuNC@BSA increase in the direction of 5-OMe-STC (*K* = 1.5 × 10^5^ M^−1^) < FB_1_ (*K* = 2.0 × 10^5^ M^−1^) < STC (*K* = 7.3 × 10^5^ M^−1^) < AFB_1_ (*K* = 43.2 × 10^5^ M^−1^). When compared with the previously published results for mycotoxin/serum albumin binding constants, our results are somewhat different, but we believe that the observed discrepancies are due to the comparative advantages of our method, as described above, thus our results being more accurate than the published ones.

It is also shown that BSA in the AuNC@BSA cluster fully retained its properties even after reducing the gold precursor during the synthesis of the nanocluster. To prove that binding constants of the AuNC@BSA with two anti-inflammatory drugs, MEL and DSS, which were previously known to bind well to BSA, were determined. The obtained results of *K* = 3.7 × 10^5^ M^−1^ for MEL/AuNC@BSA and *K* = 6.3 × 10^5^ M^−1^ for a 1:1 DSS/BSA stoichiometry of a two-step binding process are in excellent agreement with previously published data for drugs/BSA interaction. This newly presented approach using red-fluorescent AuNC@BSA as the fast, inexpensive, and simple sensor for mycotoxins offers significant advantages over currently available mycotoxin detection techniques.

Regarding the possible application scenarios of the here presented results, many mycotoxin-contaminated samples are of complex organic composition (e.g., food); thus, careful checking of any particular sample is necessary to ensure that our AuNC@BSA does not extract any other component and give a false-positive result. Quite likely, extraction, clean-up procedures, and separation procedures are inevitable prior to AuNC@BSA-detection of the mycotoxins. However, for much simpler inorganic samples (mycotoxin-contaminated dust), the application would be much more straightforward. The most important, AuNC@BSA, can be easily chemically modified to improve their selectivity. 

## Data Availability

The data presented in this study are available on request from the corresponding author.

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
