# Peer review of "Photoluminescent Gold/BSA Nanoclusters (AuNC@BSA) as Sensors for Red-Fluorescence Detection of Mycotoxins"

_materials, 2022, doi:10.3390/ma15238448_

Round 1

Reviewer 1 Report

Sensors for red-fluorescence detection is one of the research hotpots in biomedical science field. The manuscript investigates red-fluorescence detection properties based on photoluminescent gold/BSA Nanoclusters (AuNC@BSA). After reviewing the manuscript, a minor revision needs to be given out, and there are some issues that need to be addressed as follow.

1. In the Introduction section of the manuscript, why choose the Au nanoparticles instead of Ag nanoparticles? As we all know, Ag nanoparticles maybe have wider application in biological field.

2. Figure 1a showed TEM images of Au nanoparticles, please add SAED characterization to identify Au nanoparticles.

3. Scheme 1 showed molecular characteristics of BSA, please add measurement details. If this molecular characteristics of BSA was cited from other articles, please provide relevant literature into References.

4. (i) Full name of BSA occurred in the Abstract section instead of the Introduction section. (ii) Line 33, human serum albumin (HSA, BSA)” should be modified as “human serum albumin (HSA)”. (iii) References 33, a of title a Program... should be capital letter.

5. Please provide uniform format in References section of the manuscript, i.e. 22 and 23, 31 and 32. Inconsistent format occurred in the References section.

6. In order to meet the aims of submitted journals, the numbers of references need to be added, especially the targeted Materials.

Author Response

Q1. In the Introduction section of the manuscript, why choose the Au nanoparticles instead of Ag nanoparticles? As we all know, Ag nanoparticles maybe have wider application in biological field.

A1: Yes, it is true that Ag nanoparticles have wider applications in the biological field, especially because of their previously known antibacterial properties. However, silver nanoparticles tend to oxidize which affects their stability. But in our experiments, we do not use nanoparticles (NPs) at all, we use Au-clusters of about 20-25 Au atoms, embedded in BSA. These Au-nanoclusters(AuNCs) we are using, differ from nanoparticles (NPs) in the size (NPs>10-50nm, Au-nanoclusters<<1nm), which strongly affects AuNC optical properties. For example, gold nanoparticles(NPs) with d ~10 nm are red coloured, with maximum absorbance wavelength at approximately 532 nm and have no fluorescence at all. On the other hand, AuNC are strongly red-fluorescent, which we use for mycotoxin detection. The point of our work is to take advantage of the excitation of AuNC@BSA at 540 nm to avoid any interference of absorption contribution of organic molecules on emission, leading to false results.

Q2. Figure 1a showed TEM images of Au nanoparticles, please add SAED characterization to identify Au nanoparticles.

A2: The SAED measurements of our sample would be extremely challenging to do due to their radiation sensitivity and complex structure. First, our nanoclusters are very small (~ 2 nm) and second, they are coated with BSA which makes it extremely difficult to make even a high - resolution TEM measurements. Thus, for such BSA-imbedded Au-clusters of 20-atoms no SAED measurements are reported till now.

Q3. Scheme 1 showed molecular characteristics of BSA, please add measurement details. If this molecular characteristics of BSA was cited from other articles, please provide relevant literature into References.

A3: Molecular characteristics of BSA were taken from another article and it is cited now, thank you for noticing.

Q4. (i) Full name of BSA occurred in the Abstract section instead of the Introduction section. (ii) Line 33, “human serum albumin (HSA, BSA)” should be modified as “human serum albumin (HSA)”. (iii) References 33, “a” of title “a Program...” should be capital letter.

A4: Corrected as suggested.

Q5. Please provide uniform format in References section of the manuscript, i.e. 22 and 23, 31 and 32. Inconsistent format occurred in the References section.

A5: Thank you for noticing, the literature is now consistent.

Q6. In order to meet the aims of submitted journals, the numbers of references need to be added, especially the targeted Materials.

A6: We added additional 5 references, 4 of them from Materials.

  1. Tarhini, M.; Benlyamani, I.; Hamdani, S.; Agusti, G.; Fessi, H.; Greige-Gerges, H.; Bentaher, A.; Elaissari, A. Protein-Based Nanoparticle Preparation via Nanoprecipitation Method. Materials, 2018, 11, 394. https://doi.org/10.3390/ma11030394.
  2. Masikini, M.; Williams, A.; Sunday, C.; Waryo, T.; Nxusani, E.; Wilson, L.; Qakala, S.; Bilibana, M.; Douman, S.; Jonnas, A.; Baker, P.G.L.; Iwuoha, E.I. Label Free Poly(2,5-Dimethoxyaniline)–Multi-Walled Carbon Nanotubes Impedimetric Immunosensor for Fumonisin B1 Detection., 2016, Materials, 273. https://doi.org/10.3390/ma9040273.

       29.Moon, J.;  m, G.; Lee, S. A Gold Nanoparticle and Aflatoxin B1-BSA Conjugates Based Lateral Flow Assay Method for the Analysis of Aflatoxin B1. Materials, 2012, 5, 634–43. https://doi.org/10.3390/ma5040634.

  1. Majorek, K.A.; Porebski, P.J.; Daval, A.; Zimmerman, M.D.; Jablonska, K.; Stewart, A.J.; Chruszcz, M.; Minor, W. Structural and immunologic characterization of bovine, horse and rabbit serum albumins. Mol. Immunol. 2012, 52, 174 –182. doi: 10.1016/j.molimm.2012.05.011
  2. Opálková Šišková, A.; Kozma, E.; Opálek, A.; Kroneková, Z.; Kleinová, A.; Nagy, Š.; Kronek, J.; Rydz, J.; Eckstein Andicsová, A. Diclofenac Embedded in Silk Fibroin Fibers as a Drug Delivery System, Materials, 2020, 13, 3580. https://doi.org/10.3390/ma13163580

Reviewer 2 Report

The authors developed red-fluorescent BSA encapsulated AuNCs (AuNC@BSA) for the detection of natural mycotoxins. The binding properties of BSA in AuNCs were analyzed through spectrophotometric study using two model molecules, MEL and DSS. In order to test the detection ability of AuNCs@BSA, the authors used a series of mycotoxin derivatives, and their binding constants were determined by fluorescence quenching at 700 nm following their excitation at 540 nm. The manuscript is clearly written. The quality and potential applications of this work justify publication in Materials after addressing the following requests:

1. Figure 1 is blurry and difficult to read.

2. In the Materials and Methods section 2.3, the procedures were not described in sufficient detail. I think it could be important for reproducibility to provide more details.

3. How about the stability of the AuNCs@BSA? Can the authors comment on this?

4. The dynamic light scattering (DLS) data of AuNCs and AuNCs@BSA should be added to study the hydrodynamic diameter of nanoclusters before and after encapsulation.

5. How did the authors calculate the binding constants?

6. In page 5, line 163, the authors mentioned that “AuNC@BSA show two characteristic emission peaks at 431 nm and 680 nm”. However, I was unable to find the data indicating that the emission peak occurred at 431 nm.

Author Response

Q1. Figure 1 is blurry and difficult to read.

A1: We tried to improve the quality and contrast of the pictures, we hope it looks better now.

Q2. In the Materials and Methods section 2.3, the procedures were not described in sufficient detail. I think it could be important for reproducibility to provide more details.

A2: As requested we re-wrote the procedure in a straightforward way.

Q3. How about the stability of the AuNCs@BSA? Can the authors comment on this?

       A3: The spectroscopic properties of AuNC@BSA nanoclusters were checked periodically within several months and were unchanged if AuNC@BSA solution was kept in the dark and in the fridge. According to the Yan et al. (Yan, L.; Cai,Y.; Zheng, B.; Yuan, H.; Guo, Y.; Xiao, D., Choi; M.M.F. Microwave-Assisted Synthesis of BSA-Stabilized and HSA-Protected Gold Nanoclusters with Red Emission. J. Mater. Chem., 2012, 22, 1000–1005. https://doi.org/10.1039/C1JM13457D.), the zeta potential of as-prepared AuNC@BSA is -19.2 mV at pH = 10, which indicates that this solution has good stability. It is important to say that all experiments were done within less than one month of the nanocluster synthesis.

Q4. The dynamic light scattering (DLS) data of AuNCs and AuNCs@BSA should be added to study the hydrodynamic diameter of nanoclusters before and after encapsulation.

A4: That would be impossible since BSA is the reactant in the formation of gold nanoclusters, i.e. BSA both reduces and sequesters gold atoms from the HAuCl4 precursor in situ. We cannot separate so protein-imbedded Au-nanoclusters from BSA molecules, and the DLS measurements of only AuNC@BSA would be pointless since there is no data to compare them with.

Q5. How did the authors calculate the binding constants?

A5: Titrations were performed by adding aliquots of stock solutions of studied small molecules to AuNC@BSA and monitoring emission change of AuNC@BSA caused by noncovalent binding of the small molecule. Such titration data were analysed by non-linear fitting procedure, e.g. multivariate analysis using the Specfit program [38] for various possible stoichiometries of complexes which could be formed.

Q6. In page 5, line 163, the authors mentioned that “AuNC@BSA show two characteristic emission peaks at 431 nm and 680 nm”. However, I was unable to find the data indicating that the emission peak occurred at 431 nm.

A6: We focused our data on a more useful 680 nm peak (we use it in most experiments because of its red-shift not overlapping with some bio-molecules), but as requested, now we inserted a picture where the peak at 431 nm is also shown.

Reviewer 3 Report

As mentioned by the authors in Line 16-19, “Such red emission detection is advantageous over current detection strategies for the same mycotoxins, based on complex mass spectrometry procedures or eventually by fluorescence detection in the UV range (<400 nm). ” Could the method studied in this work omit the above two steps?  

What are the specific scenarios that can be applied in the future? The author also mentioned that the sensor studied in this paper can be available for point-of-care applications in the future, then what kind of samples should be targeted and specifically implemented?

The toxin samples used in this paper are all standard samples with high purity. What are the effects on the detection accuracy and specificity of mycotoxins in natural samples?

The authors are suggested to  add corresponding application scenario prospect or necessary discussion to the above problems.

Author Response

Q1. As mentioned by the authors in Line 16-19, “Such red emission detection is advantageous over current detection strategies for the same mycotoxins, based on complex mass spectrometry procedures or eventually by fluorescence detection in the UV range (<400 nm).” Could the method studied in this work omit the above two steps? 

A1: Yes, it should be possible to do it with real samples, after carefully checking for any particular sample that our AuNC@BSA does not extract any other component and give a false-positive result. Particularly for inorganic samples (mycotoxin-contaminated dust), it should work. Further, there is a possibility of chemical upgrading of these nanoclusters so that they become specific and selective to a certain substrate, without losing their emission properties. And this is definitely the advantage of our method compared to the previously published UV Vis methods.

Q2. What are the specific scenarios that can be applied in the future? The author also mentioned that the sensor studied in this paper can be available for point-of-care applications in the future, then what kind of samples should be targeted and specifically implemented?

A2: The advantage of using nanoclusters as sensors is that they. Therefore, it is likely that in the future nanoclusters will be synthesized according to the can be chemically modified to be selective and specific to the (any) targeted substrate without losing their optical properties targeted substrate, and the fluorescence sensitivity will enable the detection of very low concentrations of the targeted substrate.

Q3. The toxin samples used in this paper are all standard samples with high purity. What are the effects on the detection accuracy and specificity of mycotoxins in natural samples?

A3: The investigation of the effect of this method on the actual samples is yet to be explored. However, as the mycotoxins occur in complex matrices including food/feed, dust or building materials in the living or in industrial settings, we suspect that extraction, clean-up procedures and separation procedure are inevitable prior to detection of the mycotoxins. However, because the emission occurs at 680 nm, plenty of interferences may still be avoided, but this is yet to be explored/confirmed.

Q4. The authors are suggested to add corresponding application scenario prospect or necessary discussion to the above problems.

A4: As requested, we added a paragraph commenting on that at the end of Conclusions.

Reviewer 4 Report

The new approach using AuNC@BSA red fluorescence reporter for mycotoxins had some advantages, still some details should be added to improve the conclusion. 

1. Formulas and R-squared values shall be provided for the standard curve drawn by Figure 4e.

2. The effect of this method used in the actual sample detection is not shown, and the detection effect in the sample mixture needs to be evaluated. Results of comparison between this test method and international standard methods should be shown.

3. The experiment of RSD needs to be added.

4. It is mentioned in conclusions part,  the experimental results are different from the reported experimental results. Please explain the specific reasons.

Author Response

Q1. The new approach using AuNC@BSA red fluorescence reporter for mycotoxins had some advantages, still some details should be added to improve the conclusion.

A1: As suggested, we added a paragraph to Conclusions commenting on the application scenario and discussing limitations and prospects.

Q2. Formulas and R-squared values shall be provided for the standard curve drawn by Figure 4e.

A2: It is written in the experimental section, but for better visibility, we also inserted the R-squared value into the Caption of Figure 4.

Q3. The effect of this method used in the actual sample detection is not shown, and the detection effect in the sample mixture needs to be evaluated. Results of the comparison between this test method and international standard methods should be shown.

A3: The investigation of the effect of this method on the actual samples is yet to be explored. However, as the mycotoxins occur in complex matrices including food/feed, dust or building materials in the living or in industrial settings, we suspect that extraction and clean-up procedures are inevitable prior to detection of the mycotoxins.

Q4. The experiment of RSD needs to be added.

A4: As requested, e added to the chapter discussing the binding of standard BSA-ligands, the following: “ All fluorimetric titration experiments were repeated several times, searching for optimal conditions, and RSD values (RSD < 10%) obtained for the binding affinity constants demonstrated that this is a sensitive and a reliable method for the study of BSA-ligand systems.”

Q5. It is mentioned in conclusions part, the experimental results are different from the reported experimental results. Please explain the specific reasons.

A5: As given in the discussion, chapter 3.2., previously reported results used methods that had to be corrected for strong absorption of light at the excitation wavelength (“inner filter effect”) – mathematical estimation which can directly influence the accuracy of results.  The advantage of our method is the excitation property of AuNC@BSA at 540 nm where most organic molecules have no absorption (including DSS, MEL and all mycotoxins). This allows us to obtain binding constants directly from fluorescent measurements, without using any mathematical corrections for the inner filter effect, thus being more reliable

Reviewer 5 Report

The research presented in the article is novel and very interesting for the audience. The authors made a quite comprehensive study. Some revisions are required.

-Lines 20-21: plese add "we" or "it was" in the sentence "here, presented a new approach.." The subject of the sentence is missing.

-In several parts of the manuscript the references are cited using roman numbers which are reported in brackets in many points of the article and should be substituted with numbers. The roman numbers should also be removed from the references list.

-Line 33: please remove BSA from the acronym of HSA

-Reference 33 is highlighted in yellow. Why?

-Lines 57 and following: why the acronyms are in bold?

-Scheme 1 is not very clear, it could be improved by moving the AuNC@BSA image to the left and put the plus sign in the center. Also the resolution of the fluorescence spectra in Scheme 1 is poor, please improve it. Also, it is not written in which solvents were these spectra in Scheme 1 acquiredand the spectrum relative to FB1 seems to present a Raman peak. Please, indicate it.

-Line 149: please specify in which way the synthetic strategy was changed in respect to the literature.

-Lines 157-158: please express the 2 concentrations in the same unit of measure

-Line 163: why didn't you show also the peak at 431 nm in the spectrum in Figure 1?

- Line 147: where it says "absorption shoulder (as well as excitation spectrum)" please substitute with "(noticeable also in the excitation spectrum).

-Paragraph 3.2: I would rewrite here what MEL and DSS stand for, to hel readers.

-Line 184 (caption of the Figure): what does uMin a cuvette mean? Please correct

-Fig. 2b, please substitutes commas with dots in the numbers on the x axis

-Caption of Fig. 3, lines 199-200: maybe it is better to say "all slits: 20", right now it is strange to have written "20 20"

-Line 211: please correct in the following way: "was obtained by the model"

-Figure 3: Why with DSS, 2 excitation wavelengths were checked and only one for MEL?

- Why for STC the control was performed with HSA? In Table 1 it is said it was with BSA as well, but in the manuscript text is written HSA.

-Please remove the journal template sentence from the acknowlwedgements

Author Response

Q1. Lines 20-21: plese add "we" or "it was" in the sentence "here, presented a new approach.." The subject of the sentence is missing.

A1: Thank you for noticing, we corrected the sentence.

Q2. In several parts of the manuscript the references are cited using roman numbers which are reported in brackets in many points of the article and should be substituted with numbers. The roman numbers should also be removed from the references list.

A2: Yes, thank you for noticing, we corrected that.

Q3. Line 33: please remove BSA from the acronym of HAS

A3: Thank you, we did that.

Q4. Reference 33 is highlighted in yellow. Why?

A4: Thank you for noticing, we removed the highlighted part.

Q5. Lines 57 and following: why the acronyms are in bold?

A5:  believe that it will be easier for the reader to follow the text.

Q6. Scheme 1 is not very clear, it could be improved by moving the AuNC@BSA image to the left and put the plus sign in the center. Also the resolution of the fluorescence spectra in Scheme 1 is poor, please improve it. Also, it is not written in which solvents were these spectra in Scheme 1 acquired and the spectrum relative to FB1 seems to present a Raman peak. Please, indicate it.

 A6: We modified Scheme 1, we hope that now it is clearer to read and we added the requested information. The lowest 3 lines in the fluorescence spectrum of AuNC@BSA when titrated with FB1 do not present a Raman peak, it is a precipitation artefact at the end of titration – we removed these last 3 lines not to confuse the reader.

Q7. Line 149: please specify in which way the synthetic strategy was changed in respect to the literature.

A7: In the original protocol of Xie et al. (Xie, J.; Zheng, Y.; Ying, J.Y. Protein-Directed Synthesis of Highly Fluorescent Gold Nanoclusters. J. Am. Chem. Soc. 2009, 131, 888–89. https://doi.org/10.1021/ja806804u), the reaction lasted for 12 hours, and was conducted at 37oC. However, we noticed that our reaction is not finished after 12 hours (probably due to the different quality of the BSA, i.e. its reducing power), so we slightly raised the temperature (to 40 oC) and prolonged the reaction time to 24 hours. Also, an important factor was the time of mixing gold precursor and BSA before adding NaOH. In our case, the mixing lasted 5 minutes, and in the original paper, this lasted for 2 minutes.

Q8. Lines 157-158: please express the 2 concentrations in the same unit of measure

A8: Done as requested.

Q9. Line 163: why didn't you show also the peak at 431 nm in the spectrum in Figure 1?

A9: We focused our data on a more useful 680 nm peak (we use it in most experiments because of its red-shift not overlapping with some bio-molecules), but as requested, now we inserted a picture where the peak at 431 nm is also shown.

Q10. Line 147: where it says "absorption shoulder (as well as excitation spectrum)" please substitute with "(noticeable also in the excitation spectrum).

A10: We changed the text.

Q11. Paragraph 3.2: I would rewrite here what MEL and DSS stand for, to help readers.

A11: we added full names in the headline.

Q12. Line 184 (caption of the Figure): what does uMin a cuvette mean? Please correct

A12: Thank you for noticing, a space was missing, it means “mM in a cuvette”.

Q13. Fig. 2b, please substitutes commas with dots in the numbers on the x axis

A13: We changed commas with dots, thank you for noticing.

Q14. Caption of Fig. 3, lines 199-200: maybe it is better to say "all slits: 20", right now it is strange to have written "20 20"

A14: We changed it into the appropriate format of writing, thank you for noticing.

Q15. Line 211: please correct in the following way: "was obtained by the model"

A15: We changed the text, thank you for noticing.

Q16. Figure 3: Why with DSS, 2 excitation wavelengths were checked and only one for MEL?

A16: Because of the inner filter effect. AuNC@BSA could be excited at two wavelengths: at 365 nm and 540 nm. If one looks at the absorption spectrum of DSS, he can see that DSS does not absorb at any excitation wavelength of AuNC@BSA and that makes DSS perfect for determination of the validity of the method at two exc. wavelengths. However, MEL absorbs at 362 nm (with quite a high molar absorption coefficient) which overlaps the 1st. excitation wavelength of AuNC@BSA (365 nm) and MEL absorbance could cause an inner filter effect giving inaccurate results – thus only 540 nm excitation was used.

Q17. Why for STC the control was performed with HSA? In Table 1 it is said it was with BSA as well, but in the manuscript text is written HSA.

A17. In general, HSA and BSA should bind small molecules very similarly. Since in the literature HSA/AFB1 binding constant as reported (ref 39, commented below Table 1), but no data were known for STC, we for comparison reasons with aforesaid literature data performed titration of STC with HSA.

Q18. Please remove the journal template sentence from the acknowledgements

A18: We removed the sentence, thank you for noticing.

Round 2

Reviewer 4 Report

The English writing is good, the references cited are reasonable, the experimental results and the results in supplementary materials can support the experimental conclusions, the experimental design is rigorous, and the experimental data are authentic and reliable.